# The Role of Immunotherapy in a Tolerogenic Environment: Current and Future Perspectives for Hepatocellular Carcinoma

**DOI:** 10.3390/cells10081909

**Published:** 2021-07-27

**Authors:** Liliana Montella, Federica Sarno, Annamaria Ambrosino, Sergio Facchini, Maria D’Antò, Maria Maddalena Laterza, Morena Fasano, Ermelinda Quarata, Raffaele Angelo Nicola Ranucci, Lucia Altucci, Massimiliano Berretta, Gaetano Facchini

**Affiliations:** 1ASL NA2 NORD, Oncology Operative Unit, “Santa Maria delle Grazie” Hospital, 80078 Pozzuoli, Italy; mariamaddalena.laterza@aslnapoli2nord.it (M.M.L.); ermelinda.quarata@aslnapoli2nord.it (E.Q.); 2Precision Medicine Department, “Luigi Vanvitelli” University of Campania, 80138 Naples, Italy; federica.sarno@unicampania.it (F.S.); lucia.altucci@unicampania.it (L.A.); 3ASL NA2 NORD, Internal Medicine Operative Unit, “Santa Maria delle Grazie” Hospital, 80078 Pozzuoli, Italy; annamaria.ambrosino@aslnapoli2nord.it (A.A.); maria.danto@aslnapoli2nord.it (M.D.); raffaeleangelo.ranucci@aslnapoli2nord.it (R.A.N.R.); 4Department of Precision Medicine, Division of Medical Oncology, “Luigi Vanvitelli” University of Campania, 80131 Naples, Italy; sergio.facchini@studenti.unicampania.it (S.F.); morena.fasano@unicampania.it (M.F.); 5Department of Clinical and Experimental Medicine, University of Messina, 98121 Messina, Italy; mberretta@unime.it

**Keywords:** hepatocellular carcinoma, immunotherapy, immune checkpoint inhibitor, pathogenesis

## Abstract

**Simple Summary:**

The liver can be considered an immune organ, given its role as a hub for gut-derived antigens and liver-resident immune cells and the tolerance status of its environment. However, chronic inflammation represents a disruption to this balance and, pathogenetically, represents the beginning of a multistep process leading to cancer. The present study aimed to describe the key points of liver cancer pathogenesis, which may help in understanding the limits and perspectives of investigations concerning immunotherapy.

**Abstract:**

In contrast to several tumors whose prognoses are radically affected by novel immunotherapeutic approaches and/or targeted therapies, the outcomes of advanced hepatocellular carcinoma (HCC) remain poor. The underlying cirrhosis that is frequently associated with it complicates medical treatment and often determines survival. The landscape of HCC treatment had included sorafenib as the only drug available for ten years, until 2018, when lenvatinib was approved for treatment. The second-line systemic treatments available for hepatocellular carcinoma include regorafenib, cabozantinib, ramucirumab, and, more recently, immune checkpoint inhibitors. However, the median survival remains below 15 months. The results obtained in clinics should be interpreted whilst considering the peculiar role of the liver as an immune organ. A healthy liver microenvironment ordinarily experiences stimulation by gut-derived antigens. This setup elucidates the response to chronic inflammation and the altered balance between tolerance and immune response in HCC development. This paper provides an overview of the mechanisms involved in HCC pathogenesis, with a special focus on the immune implications, along with current and future clinical perspectives.

## 1. Introduction

Hepatocellular carcinoma (HCC) is most widespread in Africa and Asia, particularly in China, which accounts for more than 50% of the total HCC patients in the world [1]. In Europe, HCC ranked as the third leading cause of cancer-related death in 2012 [1]. The rate of survival at 5 years remains around 20%. These factors make it a serious public health problem.

Despite remarkable advances in preventive measures, including HBV vaccines and effective antiviral drugs, as well as improvements in diagnosis and management, only 30–40% of HCC patients are eligible for potentially curative therapies, which include surgical resection, transplantation, and percutaneous ablation. Most patients present with advanced disease at diagnosis or show recurrences even after potentially curative treatments. Sometimes, a decompensated liver limits the scope for medical treatment. HCC in cirrhosis has worse outcomes than HCC in healthy livers [2]. The management of cirrhosis has hardly improved over time, and its progression remains difficult to control.

The timeline of the development of HCC treatments is delineated by tyrosine kinase inhibitors [3]. In 2008, sorafenib became a milestone in the treatment of advanced/metastatic HCC when it was used as a first-line therapy [4]. However, the survival improvement was only 2.8 months compared to placebo, and the median survival was less than a year. Ten years after this first redefining therapy for advanced primary HCC, another multikinase, lenvatinib, took the place of sorafenib, given the non-inferiority of its results [5]. This drug is characterized by a high response rate, which is important when tumor shrinkage is required and can be useful in multimodal strategies.

As regards second-line treatment, two tyrosine kinase inhibitors, regorafenib and cabozantinib, and one monoclonal antibody, ramucirumab, are FDA-approved and recognized by the EMA with level I evidence (evidence from at least one large randomized, controlled trial of good methodological quality, or meta-analyses showing well-conducted randomized trials without heterogeneity) and grade A for recommendability (strong evidence for efficacy with a substantial clinical benefit; strongly recommended).

In the 1990s, the mainstream of immunotherapy became interested in advanced HCC [6]. However, the results of randomized trials with cytokines did not encourage further research [7].

Immune checkpoints are key molecules expressed by lymphoid cells that, through interaction with their cognate receptors, have an inhibitory effect and thereby prevent excessive, potentially dangerous immune responses and reduce the risk of autoimmune reactions. Immune checkpoint inhibitors (ICIs) block the pathways that inhibit immune-cell activation and stimulate immune responses against tumor cells. Among the many checkpoint receptors of immune cells, cytotoxic T-lymphocyte-associated protein-4 (CTLA-4), PD-1, T-cell immunoreceptor with immunoglobulin and ITIM domain (TIGIT), T-cell immunoglobulin-3 (TIM-3), and lymphocyte activation gene 3 (LAG-3) are those most commonly targeted in cancer immunotherapy. Six ICIs, i.e., one CTLA-4 blocker (ipilimumab), two PD-1 blockers (nivolumab and pembrolizumab), and three PD-L1 blockers (atezolizumab, avelumab, and durvalumab) have been approved for the treatment of different kinds of solid and hematological tumors [8]. In recent years, immune checkpoint inhibitors (ICIs) have been tested in advanced HCC. As single agents, ICIs showed beneficial effects on survival in only a minority of patients [9,10].

In June 2020, a combination of bevacizumab and atezolizumab appeared as a first-line HCC treatment for patients of stage C, as assessed using the Barcelona staging system. Bevacizumab is a recombinant monoclonal antibody that inhibits vascular endothelial growth factor A, thus blocking angiogenesis. Atezolizumab blocks the interaction of programmed cell death protein ligand 1 (PD-L1) with programmed cell death protein 1 (PD-1), thereby releasing the immune system from a crucial limitation. In a global, open-label, phase 3 trial, this combination reduced the risk of death and progression by about 40% when compared with sorafenib [11]. This combination is FDA- and EMA-approved.

Antiangiogenetic and immunotherapeutic drugs stand out in the treatment for HCC. The need to define which subtypes could benefit from each strategy and which represent the best integration is increasing. The dynamics of immune responses in a naturally tolerogenic liver tumor microenvironment are of utmost interest. A usual tolerance response, as opposed to an effective antiviral response, is critical in HCC pathogenesis, and its modulation could help in constructing future effective immune-based strategies.

## 2. Liver Cancer Pathogenesis and Immunological Tolerance as the Basis for HCC Development

The precise mechanisms underlying HCC development are still not well understood.

HBV and HCV infection are considered the main risk factors for HCC. Other factors, including aflatoxin contact, alcohol consumption, obesity, tobacco abusing, are also involved in the carcinogenesis and progression of HCC.

Both HBV and HCV show a predominant tropism to liver cells, even when HCV maintains a reservoir within other cells, such as lymphoid or epithelial cells.

Both viruses use cell machinery to replicate actively. HCV has a short +1 open-reading frame (ORF) that produces a genome product referred to as a mini core. Mutations in codons 70 and 91 of the mini core are associated with the development of HCC and lead to the increased expression of this protein [12]. Intracellular signal transduction pathways (p53–Rb, JAK–STAT, epidermal growth factor EGF-β, transforming growth factor-beta (TGF-β), and wnt–β-catenin), cellular oncogenes (such as Ras, c-Myc, and E2F1), the cell cycle, and tumor suppressor genes are affected by the viral proteins in liver cells [13].

HCV uses the host’s system for its own benefit. Epidermal growth factor receptor (EGFR) and some other proteins are involved in the entry mechanisms [14], and, in turn, HCV promotes the expression of EGFR (Figure 1) [15]. Similarly, the transcription activator STAT3 promotes HCV replication and is activated by HCV [15]. STAT3 signaling is involved in the balance between M1 and M2 macrophages, which have pro- and anti-inflammatory properties, respectively.

Genomic alterations, including somatic mutations, homozygous deletions, and amplifications in the TGF-β signaling pathway, were found in 39% of 9125 tumor samples across 33 cancer types in the Cancer Genome Atlas (TCGA) [16]. Increasing data highlight the crucial role of TGF-β in HCC. TGF-β is a versatile cytokine belonging to the TGF superfamily. It produces fibrogenic/proinflammatory, tumor-suppressive, and/or prometastatic effects [17]. TGF-β signaling appears to be altered at the transcriptomic level.

Wnt signaling is frequently hyperactivated and promotes liver tumor growth and dissemination [18]. Moreover, wnt signaling induces polarization to the M2 phenotype, switching the immune system to an anti-inflammatory status [19].

Differently from HCV, HBV integrates into cell DNA, thus promoting mutagenesis (Figure 2). This translates into an increased HCC risk under conditions of minimal fibrosis and no evidence of cirrhosis. The risk for HCC correlates with HBV viremia [20]. Otherwise, HCC in chronic HCV carriers only develops into cirrhosis progression [13].

Several cancer-relevant genes, including cyclin A, telomerase reverse transcriptase (TERT), platelet-derived growth factor receptor beta (PDGFRB), and mitogen-activated protein kinase 1 (MAPK1), change their expression following the integration of HBV into host DNA [21]. TERT promoter mutations were found to be significantly more associated with HCV infection than HBV infection [22]. Telomerase lengthens the telomeres in DNA strands and can change the fates of senescent cells, which, instead of undergoing apoptosis, can become immortal, as is the case with cancer cells. Telomerase activity is associated with the number of cell divisions and plays an important role in the immortality of cell lines, such as cancer cells. TERT promoter mutations mainly occur in tumors derived from tissues with limited regenerative potential, such as HCC and glioma [23].

HBV and HCV are indirect carcinogens that operate through the induction of chronic inflammation. Chronic activation of the immune system results in exhausted immune cells, the partial clearance of infected liver cells, and the continuous stimulation of liver regeneration, increasing the risks of genetic and epigenetic changes [13].

Inflammation induces tumor initiation in many ways—the increased production of reactive oxygen species (ROS) and reactive nitrogen intermediates that cause DNA damage and genomic instability [24]; the inactivation of mismatch repair enzymes; increases in stem cell-like populations; and the increased production of proinflammatory cytokines that activate subscription factors, as well as genes related to tumor proliferation and survival. The most important variants of HCC are strictly related to inflammation: hepatitis-, alcohol- and non-alcoholic steatohepatitis (NASH)-related HCC.

It is counterintuitive that an effective antiviral response could occur in the form of acute hepatitis, conferring risk of liver failure; therefore, the hospitable reception of the virus and establishing a balance between virus and host are perhaps the only acceptable ways for the liver to preserve itself. This is a sort of Sisyphean struggle: when the immune system fails to clear out viruses, sustained cycles of necrosis–inflammation–regeneration are established [25]. In response to damage, hepatocytes proliferate, thus enabling the propagation of epigenetic alterations, oncogenic mutations, and telomere shortening with consequent genomic instability. The next step involves increasing the negative regulatory immune mechanisms that preserve tissue in response to the damage caused by this struggle.

Similar to what happens during HCV infection, HBV proteins interfere with transcription factors and inflammatory responses. They also sustain oxidative damage and contribute to the increased amounts of mutations in the host genome. Through inducing the hypo- and hypermethylation of host DNA, as well as increased histone deacetylation, HBV interferes with the expression of cellular oncogenes and tumor suppressor genes [26].

Earlier studies have shown that HCV encodes proteins that facilitate the evasion of immunological surveillance [27]. In particular, HCV NS2, NS3, NS3/4A, NS4B, and NS5A proteins are involved in this immune evasion [28].

All the adopted mechanisms contribute to infection persistence and the ultimate progression into cancer.

Recent insights into HBV and HCV hepatocarcinogenesis have shown that different epigenetic changes take place. HCV induces the upregulation of genes involved in the immune-related and defense response pathways, in particular, the HLA-A, STAT1, and OAS2 genes [29]. This poses the hypothesis that a different response to immunotherapy would depend on the virus’ pathogenesis.

Portal vein flow continuously exposes the liver to antigens, and many protective tolerogenic mechanisms have been developed. As an example, liver sinusoidal endothelial cells (LSECs), also called antigen-presenting cells (APCs), express high levels of PD-L1 and low levels of the co-stimulatory molecules CD80 and CD86 [30]. This door-like system supervises antigen trafficking and the related responses. As further examples of systems with many inhibitory features, CTLA-4 and PD-L1 have been shown to be increased in expression in chronic hepatitis B and C, respectively [31,32].

In the liver, a state of immunological tolerance is maintained through several mechanisms. Kupffer cells (KCs) are liver-resident macrophages that regulate tissue homeostasis, preserving tissue via unconsidered self-damaging attacks. They eliminate apoptotic cells and cell debris via homeostasis through a flexible expression of membrane receptors [33]. KCs produce immunosuppressive cytokines, such as IL-10 and prostaglandins [34]. Similar to LSECs, they express high levels of PD-L1. The increased expression levels of PD-L1 and galectin-9 inhibit the antitumor response through the activation of PD-L1/PD-1 and galectin-9/TIM-3 signaling in T-cells [33].

Distinct macrophages are found in the liver, which can be either monocyte-derived macrophages or recruited peritoneal macrophages. Monocyte recruitment is realized through CCL2–CCR signaling. CCL2 is a member of the chemokines, which are a family of small cytokines that induce chemotaxis in responsive cells.

During HCC development, liver macrophages produce the pro-angiogenic factors TGF-β, VEGF, and PDGF, which together promote tumor growth [33]. As previously mentioned, TGF-β is considered a master immune regulator (Figure 3) [17]. HCV infection directly interferes with TGF-β, which is ordinarily produced in the liver by LSECs and hepatic stellate cells. TGF-β plays a critical role in the balance between immune tolerance and activation. It mainly induces Th17 cells (which are proinflammatory), Th2 and Th1 cell-switching, the redirection of the immune response towards B-cell rather than macrophage and CD8+ stimulation, natural killer (NK) suppression, and the differentiation of M2-type macrophages characterized by anti-inflammatory activities. Moreover, TGF-β directly increases PD-1 expression in cancer cells [17].

A recent metanalysis showed the prognostic role of tumor-associated macrophages (TAMs) in HCC. For most of the TAMs studied, a high density correlated with poor survival [35].

Thymus-derived naturally occurring regulatory T-cells (Tregs) and myeloid-derived suppressor cells (MDSCs) cooperate in creating a protective system around tumor cells.

The leaky gut and bacterial dysbiosis associated with chronic hepatitis infection and progression also contribute to chronic inflammation. The role of the gut–liver axis in HCC is being acknowledged with increasing frequency. The gut and liver are embryologically, anatomically, and functionally linked. The gut microbiota includes a wide array of bacteria and microorganisms that have a symbiotic relationship with the host. Disease predisposition and evolution are determined by the diversity of and changes in gut microbiota. There are approximately 40,000 types of microbes in the gut, and among these, the most commonly represented are Bifidobacteria, Lactobacillus, Clostridium, and Streptococcus [36]. This population changes under chronic hepatitis, regardless of the pathogenesis. As an example, Bifidobacteria and Lactobacillus are significantly less present in patients with chronic HBV and cirrhosis, while Enterococcus and Enterobacteriaceae are increased compared to healthy subjects [36]. These latter harmful bacteria increase gut mucosal permeability and so can more easily enter the liver via portal vein flow. Intestinal pathogen-associated molecular patterns (PAMPs) induce a natural immune response, which is mediated by pattern recognition receptors (PPRs); among these are toll-like receptors (TLR). These receptors are usually expressed on macrophages and dendritic cells and recognize molecules derived from microbes with a preserved structure. The intestinal PAMPs associated with chronic HBV mainly include lipopolysaccharide (LPS), unmethylated CpG DNA, bacterial cell wall components, and bacterial DNA/RNA. Among these, one pathway with beneficial properties, CpG DNA–TLR9, is weakened, while another, LPS–TLR4 (with harmful properties), is upregulated during HBV infection [36]. LPS is also harmful in the gut, where it increases mucosal permeability. In the liver, LPS activates the NF–κB pathway and induces tumor necrosis factor α (TNF-α), IL-1, and IL-6, which promote liver injury and stimulate the release of immunosuppressive mediators, such as IL-10 [36]. Leaky gut, bacterial dysbiosis, microbe-associated molecular patterns, and bacterial metabolites act as key pathways in cancer-promoting liver inflammation, fibrosis, and genotoxicity, which contribute to HCC [37]. In support of this finding, extracts from the microbiota of patients with HCC along with non-alcoholic fatty liver disease were shown to specifically induce a T-cell immunosuppressive phenotype, which demarcated them from a control group [38].

*Streptococcus salivarius* was shown to be significantly enriched during liver cirrhosis and HCV-related HCC, suggesting that it plays a pivotal role in the progression of chronic hepatitis into liver cirrhosis and ultimately HCC [39,40]. *Streptococcus salivarius* downregulates innate immune responses and, therefore, may favor the progression of HCC.

The gut microbiota also plays a role in anticancer responses: a dysbiotic microbiota composition lacking immunostimulatory bacteria or containing immunosuppressive species causes treatment failure [41]. The gut microbiome was recently recognized to influence the effectiveness of PD-1-based anticancer immunotherapy, and the authors concluded that healthy gut flora is a determinant of the anticancer response [42]. The balance between species has a direct influence on the response to checkpoint-inhibitor treatment, as shown in recent studies in HCC patients [43]. All these findings indicate that imbalances in the species found in HCC patients influence HCC promotion and the response to immune therapies.

To summarize, several immunotolerance mechanisms dominate in the promotion of HCC growth. The alterations to cell machinery caused by viruses, and the prevalence of the immunosuppressive status, suggest that immune-based approaches to this tumor type are supported by a solid rationale.

### Dynamics of Immune Cells in HCC

Approaching immune cells and immune responses typically comprises a few key stages. The dynamic immune landscape was recently assessed through the characterization of more than 75,000 individual CD45+ cells derived from 16 liver cancer patients and multiple lymphoid sites [44]. This extensive study, which included transcriptome profiles, revealed a dynamic picture wherein myeloid suppressive cells develop within the tumor and interact with other immune cells, not only in their surroundings but also in the lymph nodes and in ascites. In this study, dendritic cells were suggested to lead to T-cell dysfunction rather than contribute to T-cell maturation. LAMP-3 (lysosome-associated membrane glycoprotein 3) is a protein found almost exclusively on the surfaces of mature dendritic cells. LAMP-3+ dendritic cells migrate from tumors to lymph nodes and ultimately promote the migration of T-cells to tumors that require effector T-cells. A fascinating feature of this landscape is the potential for these migratory, multidirectional-flow immune cells to condition the immune response at distant sites. In this dynamic landscape, ascites is not an inactive state but are enriched with myeloid and lymphoid cells.

The single-cell analysis of primary and relapsed HCC revealed a different immune profile in different cancer stages [45]. When compared, early recurrences showed reduced levels of regulatory T-cells, increased dendritic cells (DCs), and infiltrated CD8+ lymphocytes, compared to the primary occurrence. Such CD8+ lymphocytes are characterized by the overexpression of KLRB1 (CD161) and present in an innate-like state of low cytotoxicity, low clonal expansion, and low expression of co-stimulatory and checkpoint molecules. The presence of these KLRB1 lymphocytes correlates with a worse prognosis. As in the previous study, dendritic cells in recurrent tumors lose their boosted effector function despite their high prevalence in infiltrates, confirming the strained immune response. Relapsed HCC contains a higher proportion of PD-L1+ malignant cells than primary tumors. The CD80 on dendritic cells preferentially binds to PD-L1 rather than the CD28 on resting T-cells, which represents another means of the promotion of immune evasion.

This picture suggests a complex reality with a pathogenetic relevance that requires further investigation.

## 3. Targeting Immunosuppressive Cells in the Tumor Microenvironment

The previously described immune-suppressive role of the HCC milieu increases the hope for therapies targeted at each specific immune cell population. However, most studies are still in the preclinical phases.

### 3.1. Targeting TAMs

TAMs are primarily involved in LC. Blocking the recruitment of macrophages could be an interesting approach to HCC. In this context, there is a CCR2 antagonist that is able to block CCL2/CCR signaling named 747, and there are also antibodies targeting glypican-3 (GPC3), which is overexpressed in LC and involved in chemotaxis. Codrituzumab (GC33) is one such antibody. A randomized phase II study evaluating codrituzumab versus placebo did not produce significant results [46]; as such, current active enrolling trials are focusing on T-cells engineered to express a GPC3–chimeric antigen receptor (GLYCAR T cells).

Colony-stimulating factor-1 (CSF-1) and its receptor, CSF-1R, regulate the differentiation and function of macrophages. Small molecule inhibitors and antibodies targeting CSF-1 and CSF-1R could contribute to the re-education of macrophages. Repolarizing macrophages towards an M1 phenotype can be achieved by the inhibition of CSF-1R with the inhibitor PLX3397 [47]. In HCC mouse models, PLX3397 delays tumor growth.

Cabiralizumab is an investigational antibody that inhibits CSF-1R, and that has been shown in preclinical and clinical studies to block the activation and survival of monocytes and macrophages [47]. Based on early (pre-) clinical models, the inhibition of CSF1R reduces the number of immunosuppressive TAMs in the tumor microenvironments of several different cancers, and it also enhances the immune response against tumors. A phase II study (NCT04050462) has been evaluating cabiralizumab combined with the anti-PD1 antibody nivolumab [48].

Low-molecular-weight fucoidan (Oligo-Fucoidan) is a polysaccharide with a variety of biological effects. Oligo-Fucoidan polarizes monocytes toward M1-like macrophages and reverses the M2 phenotype into M1 [49]. A phase II study (NCT04066660) is currently testing this supplement against placebo in advanced untreated HCC [50].

YIV-906 is a botanical cancer drug that can enhance immune function in the tumor microenvironment (by polarizing M1 macrophages and activating T-cells), protect the gastrointestinal tract (by inhibiting inflammation via IL-6, NF-kappa-B, COX2, and iNOS pathways), and contribute to intestinal tissue repair through the wnt signaling pathway [51]. Given the previously cited properties, it is frequently used as an adjuvant. Recently, YIV-906 was shown to increase the therapeutic index of capecitabine in advanced HCC [52]. YIV-906 has been observed to enhance the antitumor activity of sorafenib in preclinical models of HCC and has shown promise in preliminary clinical studies of liver, pancreatic, colon, and rectal cancers. A phase II randomized placebo-controlled study of the combination of YIV-906 and sorafenib is ongoing in HBV (+) HCC (NCT04000737) [50].

TPST-1120 is a first-in-class selective PPAR (peroxisome proliferator-activated receptor) α antagonist [53]. The PPARs are a group of nuclear receptor proteins that function as transcription factors regulating the expression of genes involved in differentiation, metabolism, and cancer. TPST-1120 is designed to exert a dual action: it targets tumor cells directly and suppresses immune cells in the tumor microenvironment. In multiple animal studies, TPST-1120, when used as a monotherapy or in combination with other anticancer drugs, showed significant tumor-reducing effects and induced durable antitumor immunity. TPST-1120 is currently being used in a phase I trial as a monotherapy and in combination with Nivolumab (NCT03829436) [50].

### 3.2. Targeting MDSCs

HuMax-IL8 (now known as BMS-986253) is a fully human monoclonal antibody that inhibits interleukin-8 (IL-8), a chemokine with direct and indirect tumor-promoting effects, mediated by immune escape and the recruitment of myeloid-derived suppressor cells [54]. Its synergistic activity with other antibodies, such as nivolumab or the anti-CSF-1R cabiralizumab, in advanced HCC, is currently under evaluation in the active phase II trial NCT04050462 [50].

Sitravatinib is a potent inhibitor of several closely related RTKs, including the VEGFR2, KIT, and TAM families (TYRO3, AXL, and MER) [55]. This converts the immunosuppressive tumor microenvironment (TME) into an immune-supportive TME. In particular, sitravatinib depletes MDSCs and repolarizes macrophages towards the proinflammatory M1 phenotype. Tislelizumab (BGB-A317) is a humanized IgG4 anti-PD-1 monoclonal antibody specifically designed to inhibit binding to FcγR on macrophages [56]. In preclinical studies, this was shown to reduce the antitumor activity of PD-1 antibodies because T effector cells became the target for antibody-mediated killing by macrophages. A phase II study that is currently recruiting (NCT03941873) is investigating sitravatinib alone and in combination with tislelizumab in unresectable locally advanced or metastatic HCC [50].

CD11b plays an important role in the recruitment and biological functions of myeloid cells, in which it is highly expressed. GB1275 is a first-in-class CD11b modulator that interferes with the balance between immune-suppressive and proactive immune reactions, specifically reducing MDSCs and TAMs at the tumor site, converting M2 immunosuppressive TAMs into an M1 phenotype, and increasing the levels of activated CD8+ T-cells in preclinical tumor models [57]. GB1275, as a monotherapy and in combination with an anti-PD1 antibody, is currently under investigation in a phase II active trial enrolling specified advanced solid tumors (NCT04060342) [50].

### 3.3. Drugs in Phase I Trials

The following other drugs are currently in phase I trials: an oral STAT3 inhibitor, TTI-101; a wnt pathway Porcupine inhibitor, CGX1321; BCA101 (which is a bifunctional antibody that blocks EGF and TGF-β); and novel antibodies, such as ABBV-151 and KY1044 [50]. ABBV-151 is a first-in-class monoclonal antibody that binds to the GARP–TGF-β1 complex and blocks the release of TGF-β1. Preclinical data show that the dual targeting of both GARP–TGF-β1 and PD-1 improves antitumor effects compared with anti-PD-1 alone. KY1044 is a human monoclonal IgG1 that selectively binds to an inducible T-cell co-stimulator (ICOS), a protein with a rate of expression differentiated by cell type. KY1044 exerts antitumor activity through the preferential depletion of intratumoral regulatory T-cells and the stimulation of effector T-cells with low levels of ICOS.

## 4. Looking at Predictive Factors for Response to Immunotherapy in HCC

The choice of immunotherapy in cancer necessitates an evaluation of factors predictive of response.

The first factor to be identified was tumor-infiltrating lymphocytes (TILs). Despite their strong immunosuppressive effects within the intrahepatic space, TILs are frequently present in HCC. A meta-analysis showed that some TIL subsets represent prognostic biomarkers in HCC [58]. Moreover, high levels of intratumoral CD8+ TILs were associated with better overall survival (OS; HR = 0.676, *p* = 0.001) and disease-free survival (disease-free survival (DFS); HR = 0.712, *p* = 0.002) [59].

The mechanisms of tumor immune suppression include increased expression of PD-L1. The binding of PD-L1 to the inhibitory checkpoint molecule PD-1 inhibits T lymphocyte proliferation, survival, and effector functions (cytotoxicity and cytokine release), as well as inducing the apoptosis of tumor-specific T-cells and promoting the differentiation of CD4+ T-cells into Foxp3+ regulatory T-cells.

The expression of PD-L1 and its relevance for HCC patients is controversial. However, mounting evidence supports the correlation of PD-L1 expression with unfavorable tumor characteristics and poor outcomes [60,61]. High expression levels of PD-L2 on tumor membranes and PD-L1 in the immune stroma have both been shown to be significantly associated with poorer OS and DFS. Macrophages, previously described as key immune cells in the tumor microenvironment, were identified as the main immune cell subtype expressing both PD-L1 and PD-L2 [62].

Among the investigated biomarkers, those that consistently show predictive value include the tumor mutational burden (TMB), which reflects the genetic alterations in a given tumor and the related production of novel antigens. The highest levels of TMB are found in melanoma, followed by non-small-cell lung cancer (NSCLC), and other squamous carcinomas, while leukemias and pediatric tumors have shown the lowest levels of TMB [63].

The aggregate data from multiple studies on small-cell lung cancer (SCLC), NSCLC, and urothelial carcinoma approximate the threshold of beneficial enrichment of immune checkpoint inhibitors (ICIs) in high-TMB tumors to be ~200 missense mutations, which is equivalent to 10 mut/Mb in FoundationOne testing or ~7 mut/Mb in MSK-IMPACT testing [64]. A recent study inferred a TMB of about two per megabase from an HCC dataset [65], which is far below the previously cited threshold of 7–10 mut/Mb.

In a study investigating the relationship between PD-L1 expression and TMB in several tumors [66], HCC was categorized together with urothelial, renal cell, and squamous cell lung carcinoma. In this study, an unbiased regression tree algorithm identified that hypermutated tumor types with TMB > 10 have the best predicted ORR (38%), regardless of PD-L1 positivity. However, the response rates increase proportionally with PD-L1 expression in cancer types with fewer than ten mutations/Mb.

It is important to consider the proportion of neoantigens, i.e., unique peptides derived from tumor-specific mutations presented as natural HLA ligands and recognized by T-cells, required to produce an effective immune response. HCC expresses multiple tumor-associated antigens with identified immunogenicity (GPC3, AFP, SSX-2, NY-ESO-1, EpCAM, and midkine). The coexpression of these antigens favors immune cell infiltration and influences disease outcomes [67]. In order to effectively induce an immune response, a neoantigen must be presented by the HLA ligand, and the presentation must efficiently induce an immune response.

The principle of “one size does not fit all” can also apply to HCC. Immunotherapy benefits inflamed tumors differently from cold/immune-excluded or desert tumors. Recently, there have been several attempts to classify LC with therapeutic implications. In one study, three major subtypes of HCC were identified: (1) mitogenic and stem cell-like tumors with chromosomal instability; (2) CTNNB1-mutated tumors (CTNNB1 codes for β-catenin) displaying immune suppression; and (3) metabolic disease-associated tumors, which included an immunogenic subgroup characterized by macrophage infiltration and favorable prognosis [67].

In this regard, defining an immune/TGF-β signature seems to be the best therapeutic approach. Four HCC subtypes can be identified: exhausted, which corresponds to a highly activated TGF-β signature; excluded with an activated TGF-β signature; active immune with a normal TGF-β signature; excluded with an inactivated TGF-β signature [17]. In the exhausted class of HCC, checkpoint inhibitors may be used, but immunotherapy resistance is clear. Antifibrotic agents have a better chance of therapeutic effectiveness. Excluded and active immune subtypes display increased rates of response to ICI. In the excluded tumors, which have an activated TGF-β signature, the synergistic effects of combined immunotherapy plus anti-TGF-β could result in a better response. For the excluded tumors with an inactivated TGF-β signature, targeted and T-cell therapies are more promising. These latter cold tumors may particularly benefit from “supra-physiological” therapies, as defined by other authors, with reference to adoptive T-cell therapies and chimeric antigen receptor (CAR) T-cells [68].

## 5. Immune Checkpoint Inhibitors in HCC

Treatments with ICIs are only significantly beneficial in a small fraction of HCC patients [9,10].

The single-arm, open-label KEYNOTE-224 trial showed an overall response rate of 17%, with 56% of responses lasting more than 12 months [10]. In the randomized phase III trial Keynote-240, pembrolizumab in the second line, when compared to the best alternative supportive care, did not achieve significant results in terms of OS and PFS [69].

Nivolumab’s approval was based on the CheckMate 040 study, a phase I/II dose-escalation and expansion trial [9]. Nivolumab displayed a manageable safety profile, with a durable objective response (15% in the dose-escalation study) [9].

These reports led to the FDA approval of using pembrolizumab and nivolumab for pretreated HCC in 2019.

In 2020, the combination of nivolumab plus ipilimumab received FDA approval based on the results of the CheckMate 040 randomized trial, which showed a significant objective response rate and durable responses [70]. A lower expression of PD-1 on T-cells was posed as a possible explanation for the low activity of the anti-PD1 antibody when used as a single agent, while the increased expression of CTLA-4 explains the greater activity of the anti-PD1 and anti-CTLA-4 combination [45]. The differential expression of immune checkpoint molecules may also explain the different effects of immunotherapy on primary and relapsed tumors [45]. After the presentation of the results of IMbrave 150, atezolizumab plus bevacizumab received FDA approval in 2020 for untreated HCC [10]. This combination was ranked IA (I: Evidence from at least one large, randomized study; A: strongly recommended) via the ESMO’s updated recommendations but has still not received EMA approval.

A systematic review of the use of ICIs in 2402 patients with advanced-stage HCC reported an objective response rate and disease control rate of 22.7% and 60.7%, respectively, and mean overall survival of 15.8 months [71].

The ongoing phase III studies with ICIs are summarized in Table 1.

Many phase III studies have focused on novel anti-PD-1 (CS1003, sintilimab, toripalimab, SCT-I10A, and camrelizumab) and novel anti-CTLA4 (IBI310) therapies. Phase II studies often involve a novel combination of ICIs and bispecific antibodies. These latter studies include tebotelimab (previously known as MGD013), which is an investigational, bispecific DART (dual-affinity re-targeting antibody) molecule designed to independently or coordinately block PD-1 and LAG-3 checkpoint molecules. Tebotelimab has been engineered to bind PD-1 and LAG-3 concomitantly or independently and disrupt these inhibitory pathways to restore exhausted T-cell function. The novel ICIs used in phase II studies include TSR-022, an anti-TIM-3 antibody, and KY1044, which selectively binds to ICOS. NKTR-214 is also named bempegaldesleukin and targets the CD122-specific receptors found on the surfaces of CD8+ effector T-cells and NK cells, stimulating the immune response. It is being investigated in combination with pembrolizumab in a phase I/II study, NCT03138889 (PROPEL) [50].

Several drugs are currently under investigation in phase I studies. These include antibodies with dual targets, such as XmAb20717, which targets PD-1 and CTLA-4; XmAb22841, which simultaneously targets CTLA-4 and LAG-3; and XmAb23104, which targets PD-1 and ICOS. SRF388, a first-in-class fully human monoclonal antibody targeting the immunosuppressive cytokine IL-27, is being investigated in a phase I study enrolling HCC patients. SO-C101 (RLI-15) is an IL-15 superagonist formulated for subcutaneous administration that is designed to be a powerful immunotherapeutic agent. It is being investigated alone and with pembrolizumab in advanced solid tumors.

## 6. Vaccines

The main cancer vaccine strategies include the dendritic cell (DC) and peptide vaccines [72]. The DC vaccine is obtained by loading DC with tumor antigens ex vivo. There is a predicted advantage provided by DC in that it promotes a tumor-specific T-cell response. However, this strategy has not produced clinically meaningful results to date.

Peptide vaccines exploit tumor-associated antigens, such as α-FP, GPC3, and TERT. Negligible clinical responses have been registered to date.

Ongoing trials point to the improved activity of vaccines obtained through novel technologies and combination strategies.

Survivin is a tumor-associated antigen that is found in tumors at much higher levels than normal [73]. DPX-Survivac (IMV^TM^) is composed of survivin-based synthetic peptide antigens combined with an adjuvant encapsulated in nanoscale lipid particles to increase its activity. This drug was tested in combination with cyclophosphamide in an immunomodulatory metronomic schedule in ovarian cancer patients. All the patients receiving the therapy showed antigen-specific immune responses [74]. An ongoing phase II trial is currently evaluating the safety and efficacy of DPX-Survivac and low-dose cyclophosphamide with pembrolizumab in selected advanced and recurrent solid tumors, including HCC (NCT03836352) [50]. Intermittent low-dose oral cyclophosphamide is used as an immunomodulator for increasing the number of survivin-specific T-cells that can be generated without inducing significant cytotoxicity.

Neoantigen vaccines are a new strategy using tumor neoantigens, which are proteins produced by tumor-mutated genes. This strategy has a preliminary multistep personalized neoantigen identification immunotherapy design, which is followed by manufacturing and treatment processes. The GT-30 study (phase 1/2 NCT04251117) is investigating GNOS-PV02 (Geneos Therapeutics), which is a personalized neoantigen vaccine delivered intradermally in combination with INO-9012 and anti-PD1 for the treatment of patients with advanced HCC [50]. INO-9012 is a DNA plasmid, coding for interleukin-12 (IL-12) [75]. While anti-PD1 is conventionally administered through intravenous injection, both GNOS-PV02 and INO-9012 are administered via skin injection, using a device called CELLECTRA 2000, which enhances the efficiency of the injection and vaccine uptake.

## 7. Adoptive Cell Therapies

Adoptive cell therapies (ACT) use immune cells, including NK cells, TILs, cytokine-induced killer cells (CIK), and CAR T-cells, to kill tumor cells [76].

Several phase II trials are ongoing [50]. The phase I/II NCT04162158 study uses allogeneic NK cells. NK cells are innate immune effectors whose antitumor activity is regulated by a large variety of inhibitory and activating receptors. Autologous NK cells directed against tumors are inhibited because they recognize self-cells through the inhibitory killer cell immunoglobulin-like receptor (KIR). On the contrary, the KIR ligand mismatch between patients and their donors, induced by allogeneic NK cells, can induce an effective antitumor response, as demonstrated in hematological malignancies.

Invariant natural killer T- (iNKT) cells, also named type I or classical NKT cells, are a distinct population of T-cells that express an invariant aβ T-cell receptor (TCR) and several cell surface molecules commonly expressed by NK cells. They are rare in the human blood pool, comprising just 0.01–1% of peripheral blood mononuclear cells (PBMCs). iNKT cells exhibit antitumor activity against malignant tumors by producing high levels of cytokines.

NCT04011033 is a phase II study combining the adoptive transfer of iNKT cells with transcatheter arterial chemoembolization (TACE) to treat advanced HCC [50]. NCT03093688 is a phase I/II study evaluating the infusion of iNKT cells and CD8+ T-cells [50].

SPEARS T-cells are specific peptide-enhanced affinity receptor T-cells that form the basis of novel research programs. ADP-A2AFP SPEAR T-cells target alpha-fetoprotein (AFP) and are under investigation in an ongoing phase I clinical trial for the treatment of patients with HCC (NCT03132792) [50].

Perhaps the most promising strategy in the ACT is the use of CAR T-cells. CAR T-cells are genetically modified T lymphocytes that specifically target tumor-associated antigens, killing tumor cells in an MHC-independent manner. A CAR combines antigen-binding and T-cell-activating functions in a single receptor. CARs are made of an extracellular antigen-binding domain, an intracellular domain, and a hinge area that enables the molecule to interact with other cells through a flexible motion. Several generations of CAR T-cells have been developed to enhance antitumor responses and limit potential side effects [76]. Fourth-generation CAR T-cells, named TRUCKs (T-cells redirected for antigen-unrestricted cytokine-initiated killing), combine the CAR’s ability to attack the tumor with the immune-modulating competence of the delivered cytokine [77]. Several phase I studies are recruiting patients with advanced HCC and are employing CAR T-cells that target GPC3, which is highly expressed in LC and correlates with a poor prognosis (NCT04121273, NCT03198546, and NCT02905188) [50].

## 8. Locoregional Plus Immunotherapy

The standard locoregional treatments used for unresectable HCC can trigger effector T-cell responses through the release of danger-associated molecular patterns (DAMPs) and may synergize with systemic immunotherapy.

Immune cell infiltration into the tumor microenvironment increases after radiotherapy (RT) because of the upregulated expression of adhesion molecules on endothelial cells and the secretion of cytokines that can recruit cytotoxic T lymphocytes [78]. By contrast, RT directly kills radiosensitive CD8 effector T lymphocytes and conversely saves the less radiosensitive Tregs [79]. The RT-induced production of TGF-β has an immunosuppressive effect. Increases in M2 macrophages and MDSCs, as well as tumor cells and T lymphocytes, enhance PD-L1 and PD-1 expression. Therefore, ICIs could potentially reverse these drawbacks. RT and CTLA-4 or PD-L1 inhibitors exhibit synergistic activity in preclinical models [80].

A series of patients with advanced HCC treated with nivolumab and radiotherapy before and/or during medical treatment had significantly longer PFS and OS compared with patients treated with RT only [81]. Better outcomes are generally observed following combined immunotherapy and RT as a local treatment, compared to other local approaches, such as surgery, radiofrequency ablation, and transarterial chemoembolization (TACE) [82].

HCC patients exhibiting macroscopic vascular tumor invasion are categorized as class C by the Barcelona staging system and are treated with sorafenib according to guidelines [83]. However, their prognosis is worse than that for patients without macrovascular invasion, as confirmed by their lower survival rate in the Sharp trial [4]. The results of combinations of TACE with sorafenib and TACE with stereotactic body radiation therapy (SBRT) favor the second approach [84]. The TACE–radiotherapy combination also achieves better results than sorafenib alone [85]. These findings have encouraged investigations into locoregional plus immunotherapy. The current actively recruiting phase III trials for immunotherapy plus SBRT and TACE are summarized in Table 2.

## 9. Discussion and Conclusions

The liver microenvironment represents a peculiar barrier to antigen stimulation, which is usual for the liver but becomes a critical factor in chronic hepatitis and subsequent tumor progression. The typing of regulatory and inhibitory cells whose receptors and ligands are key players in this setting has, to date, only elicited studies of earlier phases. However, a new era has begun for HCC.

At the ASCO Gastrointestinal Meeting held in 2021, the updated median OS following the PD-L1 inhibitor/VEGF inhibitor combination of atezolizumab plus bevacizumab first-line systemic treatment was reported to be 19.2 months, which translates into an improvement of about 5.8 months compared to that realized with sorafenib [86]. The median OS observed in this trial is longer than that in any previous phase III trial for advanced HCC and should be compared with the reference median OS achieved with sorafenib in the pivotal phase III SHARP study, which ran for just over 10 months. The results achieved with a combination of ICIs and an antiangiogenetic drug suggest that, as in other tumors, combination therapies can synergize in HCC.

A broader selection of patients deriving better results from immune therapy is needed, especially if we consider that ICIs involve the limitation of immune-related adverse events. Despite ICIs being substantially safer in differently vulnerable patients, HCC patients with underlying cirrhosis require close surveillance. Monitoring is particularly necessary when ICIs are associated with antiangiogenetic drugs, given the increased bleeding risk. The relevance of safety for HCC patients was also acknowledged in earlier studies with tyrosine kinase inhibitors, wherein a preliminary endoscopic assessment was considered essential [87,88,89]. As such, attempts to define an immune signature that can help to tailor the best treatment for each subtype are especially welcome and have recently been gaining attention [17,90,91,92].

Next-generation sequencing (NGS) studies could provide predictive and/or prognostic information for HCC patients [93]. Biological information should be incorporated into algorithms that use clinical data, such as the ECOG performance status, child function, number of lesions, volume of the targeted liver, in order to better determine the most suitable treatment.

Immune/gene HCC signatures, advancements in technologies, and significant efforts to tailor strategies of treatment define the changing landscape of HCC.

## Figures and Tables

**Figure 1 cells-10-01909-f001:**
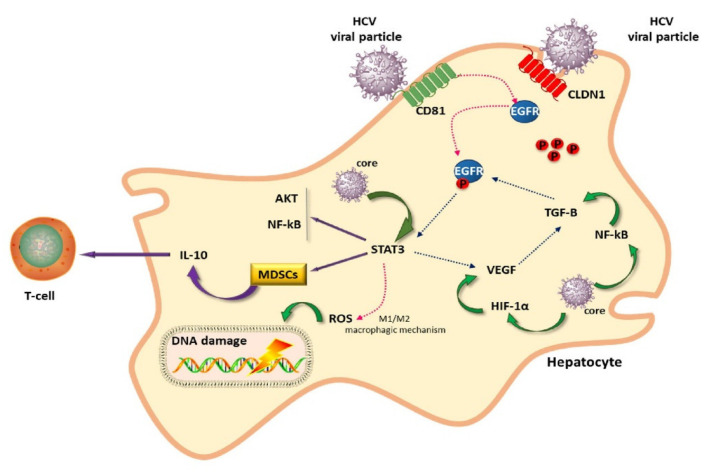
HCV liver cell infection: in this picture a representation of the complex interactions between HCV and cell machinery with involvement of intracellular pathways.

**Figure 2 cells-10-01909-f002:**
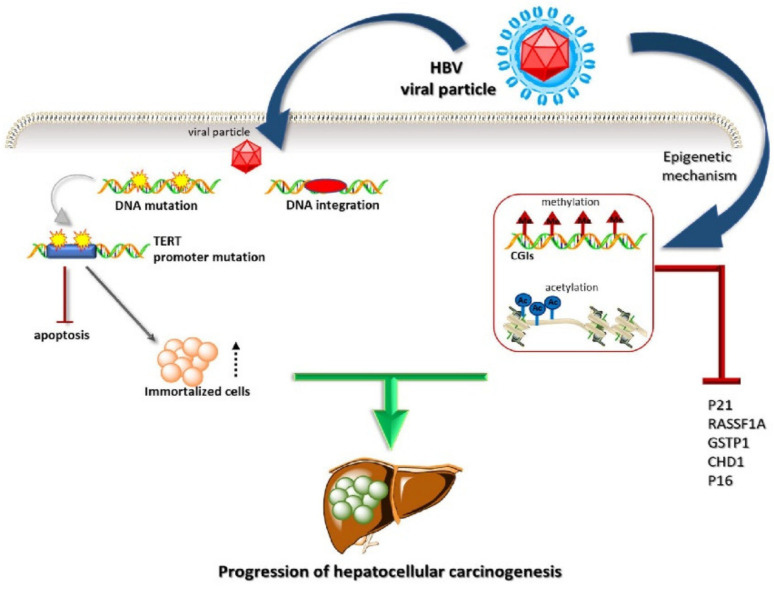
HBV liver cell infection: through integration into cell DNA, HBV changes gene expression of relevant genes and among them TERT.

**Figure 3 cells-10-01909-f003:**
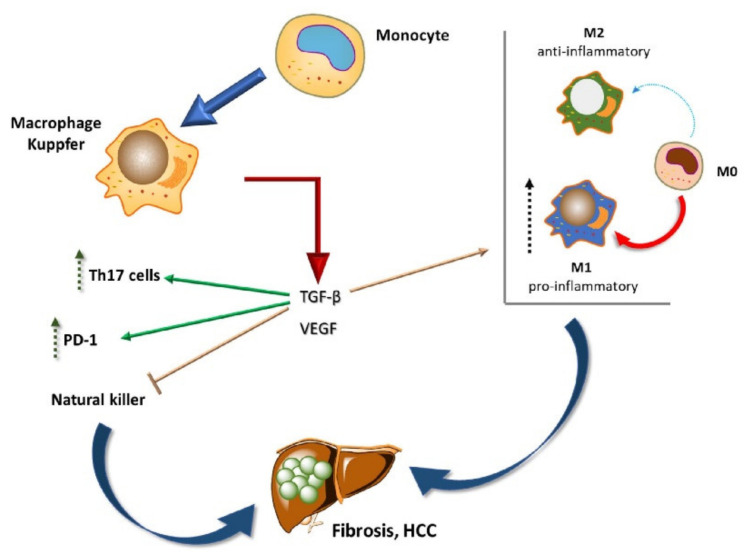
Resident macrophages are critically involved in fibrosis and HCC. The key player TGF-β rules immune tolerance and activation.

**Table 1 cells-10-01909-t001:** Active phase III and II studies with ICIs in HCC patients who have not received prior systemic therapy [50].

Trial Identifier	Drugs	Phase	Treatment Arms	Estimated Enrollment	Estimated Study Completion Date
NCT03298451 (Himalaya)	Durvalumab Tremelimumab Sorafenib	III	Durvalumab + Tremelimumab Durvalumab Sorafenib	1504 participants	30 April 2022
NCT04194775	CS1003 (anti-PD-1 antibody) Lenvatinib	III	CS1003 + Lenvatinib Placebo + Lenvatinib	525 participants	30 June 2023
CheckMate 9DW NCT04039607	Nivolumab Ipilimumab Sorafenib Lenvatinib	III	Nivolumab + Ipilimumab SOC (sorafenib or Lenvatinib)	650 participants	30 September 2023
NCT04720716	IBI310 (anti-CTLA4) Sintilimab Sorafenib	III	IBI310 + Sintilimab Sorafenib	490 participants	1 December 2023
NCT04723004	Toripalimab (anti-PD1) Bevacizumab Sorafenib	III	Toripalimab + Bevacizumab Sorafenib	280 participants	31 December 2024
NCT04523493	Toripalimab Lenvatinib	III	Toripalimab + Lenvatinib Placebo + Lenvatinib	486 participants	24 August 2024
NCT04560894	SCT-I10A (anti- PD-1) SCT510 (bevacizumab biosimilar) Sorafenib	II/III	SCT-I10A + SCT510 Sorafenib	621 participants	September 2024
NCT03605706	SHR-1210 (Camrelizumab, anti-PD-1)	III	SHR-1210 + FOLFOX4 FOLFOX4	396 participants	December 2021
NCT03755791 (COSMIC-312)	Cabozantinib Sorafenib Atezolizumab	III	Cabozantinib + Atezolizumab Sorafenib Cabozantinib	740 participants	1 December 2021
NCT04310709 (RENOBATE)	Regorafenib Nivolumab	II	Regorafenib + Nivolumab	42 participants	30 May 2023
NCT03695250	BMS-986205 (IDO1 inhibitor) Nivolumab	I/II	BMS-986205 + Nivolumab	23 participants	1 June 2022
NCT03680508	TSR-022 (cobolimab, TIM-3-binding antibody) and TSR-042 (anti PD-1 dostarlimab)	II	TSR-022 + TSR-042	42 participants	October 2023

SOC: standard of care; in phase III studies, treatment arms are indicated with bullet point letters.

**Table 2 cells-10-01909-t002:** Active phase III studies for immunotherapy plus locoregional therapies in HCC [50].

Trial Identifier	Drugs	Phase	Treatment Arms	Main Patient Characteristics	Estimated Enrollment	Estimated Study Completion Date
NCT04167293	Anti PD-1 sintilimab	III	SBRT + PD-1 arm SBRT	Portal vein invasion No previous treatment	116 participants	31 October 2022 Arms and interventions
NCT04709380	Toripalimab Sorafenib	III	Radiotherapy + Toripalimab Sorafenib	BCLC stage C with portal vein/hepatic vein tumor thrombosis	85 participants	28 February 2023
NCT04712643	Atezolizumab Bevacizumab	III	Atezolizumab + Bevacizumab + TACE TACE	No prior systemic therapy	342 participants	26 February 2027
NCT04268888 (TACE-3)	Nivolumab	III	TACE TACE + Nivolumab		522 participants	June 2026
NCT04340193 (CheckMate 74W)	Nivolumab Ipilimumab	III	Nivolumab + Ipilimumab + TACE Nivolumab + Ipilimumab-Placebo + TACE Nivolumab-Placebo + Ipilimumab-Placebo + TACE		765 participants	10 June 2028
NCT03778957 (Emerald)	Durvalumab Bevacizumab	III	TACE + Durvalumab TACE + Durvalumab + Bevacizumab TACE + Placebo		710 participants	30 August 2024
NCT04229355	Sorafenib Lenvatinib anti-PD-1	III	DEB-TACE + Sorafenib DEB-TACE + Lenvatinib DEB-TACE + PD-1 inhibitor		90 participants	30 December 2022

SBRT: stereotactic body radiation therapy; TACE: transarterial chemoembolization therapy; DEB-TACE: transarterial chemoembolization (TACE) based on drug-eluting beads; in phase III studies, treatment arms are indicated with bullet point letters.

## Data Availability

No new data were created or analyzed in this study. Data sharing is not applicable to this article.

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
