# Peer review of "The Role of Immunotherapy in a Tolerogenic Environment: Current and Future Perspectives for Hepatocellular Carcinoma"

_cells, 2021, doi:10.3390/cells10081909_

Round 1

Reviewer 1 Report

An excellent rewiev. The amount of data presented is vast and reading would be easier, if some illustrations were provided - fx of different cell types with associated molecular interactions implicated in carcinogenesis in section 2.

Author Response

Thank you for your precious help in improving our paper.

Two figures have been added with a comprehensive representation of virus-cell interactions and the third about the role of macrophages and TGF-β.

Reviewer 2 Report

This review about immunotherapy in HCC is covering a topic which has been studied extensively over the past years. Nevertheless, the authors create interesting chapters including sub-sections which cover individual immune cells, which makes this report worth reading.

The most interesting part covers the new trials studying immunsuppressive immune cells. This reviewer would appreciate some more translational aspects including data from most recent single-cell landscape studies (Zheng et al., Sun et al.) covered in an additional chapter.

Furthermore, grammar and style should be revised to cover some minor errors (e.g. Abstract: Management of frequent associated underlying cirrhosis makes..... Or 1st sentence introduction: Hepatocellular carcinoma (HCC) is .... all in China where ? accounts for more than 50% of the whole burden.) These sentences seem to be incomplete or not structured correctly. More of these examples of wrong structure or missing verbs occur throughout the manuscript. Please revise.

Author Response

Dear Reviewer,

Thank you for this precious inspiration. We have added a paragraph with the discussion on the dynamics of immune cells in HCC reporting the excellent works of Zhang Q and Sun Y (page 13).

The manuscript was revised by Cells editing service as concerns the required revisions.

We sincerely hope that the version presented could be appreciated so that you can consider our manuscript suitable for publication.

Best Regards

Reviewer 3 Report

Dear Author,

The review manuscript titled "The role of immunotherapy in a tolerogenic environment: current and future perspectives for hepatocellular carcinoma" by Montella L et al., highlighted the recent studies. I have few concerns about the tables in your review article. Please follow the directions:

The review manuscript titled "The role of immunotherapy in a tolerogenic environment: current and future perspectives for hepatocellular carcinoma" by Montella L et al., highlighted the recent studies and future perspectives of immunotherapy in hepatocelluar  immune tolerance. However, the author focused the mechanisms involved in HCC pathogenesis and its implication.

In a each section the author highlighted the pathogenesis of liver cancer, immunological tolerance, targeting associated factors responsible in a tumor microenvironment and recent update on adaptive therapies.

Overall, the content of the review manuscript is good enough. However, I have few concerns about the clinical trials. Thereby, the author has to address the few minor comments for accepting the manuscript for the publication.

Minor comments:

a). For eg. The author has mentioned NCT04167293 clinical trail in table 2. In this study there are three publications. please find the details:

  1. Shi Y, Su H, Song Y, Jiang W, Sun X, Qian W, Zhang W, Gao Y, Jin Z, Zhou J, Jin C, Zou L, Qiu L, Li W, Yang J, Hou M, Zeng S, Zhang Q, Hu J, Zhou H, Xiong Y, Liu P. Safety and activity of sintilimab in patients with relapsed or refractory classical Hodgkin lymphoma (ORIENT-1): a multicentre, single-arm, phase 2 trial. Lancet Haematol. 2019 Jan;6(1):e12-e19. doi: 10.1016/S2352-3026(18)30192
  2. Shen L, Xi M, Zhao L, Zhang X, Wang X, Huang Z, Chen Q, Zhang T, Shen J, Liu M, Huang J. Combination Therapy after TACE for Hepatocellular Carcinoma with Macroscopic Vascular Invasion: Stereotactic Body Radiotherapy versus Sorafenib. Cancers (Basel). 2018 Dec 14;10(12). pii: E516. doi: 10.3390/cancers10120516.
  3. Yoon SM, Ryoo BY, Lee SJ, Kim JH, Shin JH, An JH, Lee HC, Lim YS. Efficacy and Safety of Transarterial Chemoembolization Plus External Beam Radiotherapy vs Sorafenib in Hepatocellular Carcinoma With Macroscopic Vascular Invasion: A Randomized Clinical Trial. JAMA Oncol. 2018 May 1;4(5):661-669. doi: 10.1001/jamaoncol.2017.5847.

The author did not highlight anywhere in this review.

b) Table1 and table 2 must be clearly described (treatment arm should be mentioned clearly).

c) The review manuscript must be edited by native English speaker.

Finally, I recommend this review article for publication after the minor corrections.

Author Response

Dear Reviewer,

Thank you for your precious help in improving our paper.

We sincerely hope that the revisions performed could be appreciated so that you can consider our manuscript suitable for publication.

Following, you will find the text changes according to your suggestions.

a). For eg. The author has mentioned NCT04167293 clinical trial in table 2...... 

Thank you for your suggestion. On page 13 lines 3759-3766, we have reported the mentioned studies with related references (81-83).

b) Table 1 and table 2 must be clearly described (treatment arm should be mentioned clearly).

Tables have been revised with the explanation that treatment arms are represented with bullet point letters.

However, both tables could be clearer with a horizontal layout. 

We agreed with you. However, in the present version layout was not changed to avoid manuscript disruption. 

c) The review manuscript must be edited by native English speaker.

The manuscript will be assigned to Cells' editing service for definitive version.

Best Regards

This manuscript is a resubmission of an earlier submission. The following is a list of the peer review reports and author responses from that submission.

Round 1

Reviewer 1 Report

General comments:

  • Suggest having a native English speaker review manuscript to improve syntax and flow.  Fluency of translation is uneven throughout paper which makes it hard to follow
  • Term "liver cancer" used in some parts of the paper and HCC in others.  The focus is on HCC, not liver cancer in general, so HCC should be used uniformly throughout the paper

Specific comments:

- Abstract: Contains too much general information on general treatment landscape of HCC for a review that is supposed to be focused on immunotherapy.  The abstract as is does not clearly relate to the title of the article nor does it provider the reader with a sense of what this article is about.  

- Introduction reads like a background for a general review paper on HCC with a lot of language pertaining to the epidemiology and risk factors for HCC and overview of the various FDA approved agents instead of providing a context for the focus of the article which is on immunotherapy.  Though the authors mention bev/atezo near the end, earlier studies on nivo and pembro in HCC are not included and ought to be.  Although Section 5 does go into this, these data should be presented earlier in the paper to provide background and context for the current status of immunotherapy in HCC and where to go from here.

- Section 2.  This section is very disorganized.  It contains random sentences and paragraphs which provide a lot of details about viral hepatocarcinogenesis and various oncogenic signaling pathways, but not coherently weaving these into a narrative that explains how they promote immune tolerance and HCC progression.  It also jumps around from initially talking about hepatitis induced carcinogenesis to the gut microbiome then jumping back to viral hepatitis.  

- Section 3 could be better organized by creating subheadings for the different immune cell populations and talking about the respective therapeutic approaches in development.

- Section 4 - title is puzzling and doesn't clearly relate to the content of the ensuing paragraph which seems to discuss various predictive biomarkers of response to immunotherapy in other cancers and their presence in HCC.

- Discussion and Conclusion section is underdeveloped.  This should summarize the review and place it in context of current management of HCC and future directions.  Paragraph summarizing activity of immunotherapy in melanoma and lung cancer isn't necessary and should be deleted.

Reviewer 2 Report

The review has an interesting premise but needs to be significantly revised in structure.  There are numerous statements that are not connected and often references are missing (for example lines 139, 274, 482).

Reviewer 3 Report

Section 2 of this review provides the information in a somehow disorganized manner. This section should be redone presenting the various data with a lineal and logical connection among them and with more focus on the specific subject of the review

In Section 6 (Vaccines), and also in section 7, many statements are made without corresponding references

As the review is the work of several authors it is important to pay attention to maintain an homogenous structure from the beginning to the end of the manuscript avoiding repetitive explanations of concepts. For instance, immune check point molecules are mentioned in Section 1 but they are defined again at the beginning of Section 5.

Minor comments.

  1. In the 4th paragraph of Section 1 it is said : Numerous genetic pathways have been studied in HCC. According to the concepts mentioned in the following sentences,  it would be more appropriate to say: Numerous cell signaling pathways have been studied in HCC.
  2. No references are given in last paragraph Section 3